# Research Progress on Lipophagy-Mediated Exercise Intervention in Non-Alcoholic Fatty Liver Disease

**DOI:** 10.3390/ijms25063153

**Published:** 2024-03-09

**Authors:** Xi Li, Yangjun Yang, Yi Sun, Shuzhe Ding

**Affiliations:** 1Key Laboratory of Adolescent Health Assessment and Exercise Intervention of Ministry of Education, East China Normal University, Shanghai 200241, China; LiXi989508060506@163.com (X.L.); yjyang98@163.com (Y.Y.); 2College of Physical Education & Health, East China Normal University, Shanghai 200241, China

**Keywords:** lipophagy, exercise, non-alcoholic fatty liver

## Abstract

Lipophagy is a cellular pathway targeting the lysosomal degradation of lipid droplets, playing a role in promoting lipid turnover and renewal. Abnormal lipophagy processes can lead to the occurrence and development of non-alcoholic fatty liver disease (NAFLD), characterized by the deposition of lipid droplets (LDs) in the liver. The importance of exercise training in preventing and improving NAFLD has been well-established, but the exact mechanisms remain unclear. Recent research findings suggest that lipophagy may serve as a crucial hub for liver lipid turnover under exercise conditions. Exercise may alleviate hepatic lipid accumulation and mitigate inflammatory responses and fibrosis through lipophagy, thereby improving the onset and progression of NAFLD.

## 1. Introduction

NAFLD is the most prevalent chronic liver disease globally, affecting approximately 25% of the world’s population [1]. NAFLD is a clinical–pathological syndrome characterized by excessive fat deposition in the liver, encompassing simple steatosis (NAFL), non-alcoholic steatohepatitis (NASH), fibrosis, and even cirrhosis and hepatocellular carcinoma (HCC) [2]. Moreover, individuals with NAFLD face an increased risk of cardiovascular diseases (CVD), type 2 diabetes mellitus (T2DM), chronic kidney disease (CKD), and other conditions [3], posing a serious threat to overall health. Unfortunately, aside from lifestyle changes and weight reduction, there is currently no fully effective medical intervention to reverse NAFLD [4]. Therefore, exploring potential strategies to improve NAFLD is a pressing and crucial issue.

Abnormal lipid-droplet accumulation is a significant characteristic of NAFLD. Currently, lipid-droplet biogenesis is considered a key factor leading to this phenomenon (REF). However, besides lipid-droplet biogenesis, the homeostasis of lipid droplets also requires a balanced degradation process, a significant portion of which is achieved through autophagy. Autophagy is a cellular degradation process that clears and recycles unnecessary or damaged cellular components within lysosomes [5]. When macroautophagy selectively degrades lipid droplets, it is referred to as lipophagy. As a form of selective autophagy, lipophagy refers to the process wherein LDs in cells are transported to lysosomes for degradation during energy deficiency, acute nutrient overload, or exercise stress, promoting lipid mobilization and turnover [6]. Research has demonstrated that there is a significant accumulation of lipid droplets in the livers of patients with NAFLD or animal models thereof, indicating an impediment in the lipophagy process [7]. This obstruction in the lipophagy process further diminishes lipid mobilization within cells, reduces lipid circulation, and exacerbates abnormal lipid accumulation in the liver [6], leading to further deterioration of the disease. This underscores the critical role of lipophagy in the NAFLD process.

In the field of exercise science, moderate exercise training has been identified as an effective measure to alleviate the onset and progression of NAFLD [8]. Numerous studies have shown that aerobic endurance exercise, resistance exercise, high-intensity interval training, simulated knee-joint movement, and electrical stimulation can effectively improve NAFLD [9,10,11,12,13]. The mechanisms include enhanced mitochondrial function and increased lipid oxidation. However, the precise mechanisms are not yet fully elucidated. Recent research suggests that lipophagy may play a pivotal role in this process. Consequently, this paper comprehensively analyzes the interconnection between lipophagy, NAFLD, and exercise, elucidating potential mechanisms through which exercise, via lipophagy, may improve the onset and progression of NAFLD, offering new insights into the exercise-based prevention and treatment of NAFLD.

## 2. Lipophagy—A Cellular Autophagic Program Driving Lipid Turnover

### 2.1. Overview of Lipid Droplets

LDs are cellular organelles enveloped by a monolayer of phospholipids, storing neutral lipids such as triglycerides (TG) and cholesteryl esters (CE). They are widely distributed in various tissues and organs of organisms. The lipid composition within LDs varies among different organs; for instance, TG are the primary stored lipid in the LDs of highly metabolically active tissues like muscles, liver, and adipose, while CE predominate in the LDs of tissues like the adrenal glands, ovaries, and testes, utilized for hormone synthesis [14].

However, LDs are not merely lipid-storage depots; they are highly dynamic organelles involved in processes such as budding, fusion, expansion, and degradation, regulated by cellular lipid and energy metabolism. The dysregulation of lipid metabolism pathways, such as fatty acid intake, oxidation, TG synthesis, and TG export in the form of lipoproteins, can lead to abnormal TG accumulation in cells, disrupting LD dynamics and contributing to diseases like obesity, diabetes, and fatty liver [14].

### 2.2. Molecular Mechanisms of Lipophagy

Lipophagy is a selective form of autophagy. Specifically, lipophagy involves the transport of LDs to lysosomes under stress conditions such as energy deficiency, exercise stress, or nutritional stress. Within lysosomes, lysosomal acid lipase (LAL) breaks down LDs, generating fatty acids and other by-products for recycling. Lipophagy facilitates the breakdown of TG within cellular lipid droplets into free fatty acids (FFA), which then enter the mitochondria for β-oxidation to generate energy, thereby maintaining cellular energy demands [6]. Additionally, these FFAs may undergo resynthesis to form lipid droplets rich in TG, promoting the circulation and turnover of lipids within the cytoplasm and maintaining flexibility in lipid metabolism [15]. Alternatively, lipophagy may lead to the direct excretion of lipids into the bloodstream, facilitating information exchange with other organs [16]. Therefore, lipophagy is hypothesized to be a critical process for maintaining cellular lipid flux, supporting energy metabolism, and facilitating intercellular lipid molecular communication.

Currently, known forms of cellular autophagy include macroautophagy, microautophagy, and chaperone-mediated autophagy (CMA), and it is suggested that LD autophagy may operate through these pathways.

#### 2.2.1. Macroautophagy Pathway of Lipid Droplets

Mammalian macroautophagy is a continuous process beginning with the formation of double-membrane structures called autophagosomes. During autophagosome formation, cellular structures like misfolded/unused proteins, LDs, and mitochondria, as well as pathogens, are sequestered. Mature autophagosomes are transported by microtubules, fuse with lysosomes, and degrade their contents. This process relies on the participation of numerous autophagy-related genes (*ATGs*) encoding proteins, class III phosphatidylinositol 3-kinase (VPS34), Beclin 1 (homologous to yeast Atg6), Rab GTPases, tethering proteins, and soluble N-ethylmaleimide-sensitive factor-attachment protein receptors (SNAREs) [17].

Macrolipophagy involves the selective recognition, encapsulation, transport, and lysosomal degradation of entire or partial LDs by autophagosomes [18]. Initially observed in the livers of *mice* under nutrient-deprived conditions, researchers found that inhibiting autophagosome formation or disrupting autophagosome–lysosome fusion led to the reduced co-localization of LDs with lysosomal-associated membrane protein 1 (LAMP1), increased cytoplasmic TG, LD quantity, and reduced fatty acid β-oxidation [6]. This discovery unveiled the role of lipophagy in lipid and energy metabolism. Subsequent studies found the significance of macrolipophagy in various cell types, including neuronal cells, endothelial cells, and adipocytes [19].

Similar to macroautophagy, macrolipophagy involves numerous ATG proteins and the highly conserved core autophagy machinery. Autophagy recognition receptors such as p62/SQSTM1, NDP52, and NBR1 also play crucial roles in LD recognition [18,20]. Mass spectrometry analysis further indicates the close association of LD structural proteins, metabolic enzymes, neutral lipases, and ubiquitinated proteins with lipophagy [21], suggesting that macrolipophagy is a complex and dynamic process. Current research suggests the existence of both ubiquitin-dependent and ubiquitin-independent pathways in macrolipophagy (as shown in Figure 1).

##### Ubiquitin-Dependent Macrolipophagy

Ubiquitin (Ub) is a signaling molecule discovered in the last century that mediates the ubiquitylation modification of target proteins through the ubiquitin-activating enzyme (E1), the ubiquitin-conjugating enzyme (E2), and ubiquitin ligase (E3), or undergoes de-ubiquitination through de-ubiquitinating enzymes. It extensively participates in cellular processes such as autophagy, proliferation, apoptosis, and inflammation. Currently, research on ubiquitin-dependent macrolipophagy is relatively abundant. However, the mechanism of how lipid droplets are recognized and degraded through ubiquitination is not clear. The following will provide an overview of the potential mechanisms involved.

p62, the earliest discovered autophagy receptor protein, binds to ubiquitylated substrate proteins through its ubiquitin-associated domain (UBA domain) and targets ubiquitinated substrates to autophagic sites for degradation through its LC3-interaction region (LIR). SPART (spartin) is a crucial protein that promotes LD ubiquitination. Under conditions of oleic acid exposure, the C-terminal domain of SPART binds to the lipid-droplet membrane protein PLIN3, serving as an adaptor protein to recruit a ubiquitin ligase homologous to the C-terminus of E6-AP.The ubiquitin ligase ITCH/AIP4 (itchy E3 ubiquitin protein) can be activated and recruited to LD, potentially inducing the polyubiquitination of PLIN2 [22], facilitating the binding of PLIN2 to the autophagy receptor p62 [23], and subsequently initiating ubiquitin-dependent macrolipophagy. However, more evidence is needed to support this viewpoint. Additionally, under ethanol exposure conditions, the loss of another protein, PLIN1, on the LD membrane weakens the interaction between LC3 and p62 with LD, suggesting that PLIN1 may also be one of the proteins recognized by macrolipophagy receptors [24]. Furthermore, the Huntingtin (Htt) protein can interact with p62, facilitating the connection between p62 and LC3 and substrates marked with lys63 ubiquitin. The absence of Htt results in a reduction of phagosomes around lipid droplets, suggesting that Htt may play a crucial role in the process of lipophagy [25]. Moreover, Ancient Ubiquitous Protein 1 (AUP1), a highly conserved protein widely expressed on the LD surface, recruits and binds to the E2 ubiquitin-conjugating enzyme UBE2G2, providing a molecular link for the ubiquitination of LD and establishing a possibility for subsequent ubiquitin-dependent autophagic degradation [26], though the mechanism remains unclear.

##### Ubiquitin-Independent Macrolipophagy

In recent years, researchers have gradually identified several specific receptors for lipid-droplet autophagy. Oxysterol-binding protein (OSBP)-related protein 8, ORP8, is the first identified specific autophagy receptor located on the surface of lipid droplets. Under starvation conditions, ORP8 can be phosphorylated and modified by AMPK, enhancing the binding of autophagy to LC3/GABARAP, thereby degrading lipid droplets. This is essential and necessary for improving liver lipid-deposition-related diseases [27]. ATG14 is the core unit of the PI3KC3-C1 complex, which can be localized to the lipid-droplet membrane through its own BATS domain and recruit LC3II through a classic LIR motif, “WxxL”, initiating the autophagic degradation of lipid droplets [28]. Additionally, SPART (spartin), mainly mediates the autophagic degradation of lipid droplets in neurons, the gene mutations of which can lead to Troyer syndrome, a form of complex hereditary spastic paraplegia [29].

Patatin-like phospholipase domain-containing 2 (PNPLA2/ATGL), hormone-sensitive triglyceride lipase (LIPE/HSL) may be closely associated with ubiquitin-independent macrolipophagy processes. The results of immunoprecipitation indicate that adipose triglyceride lipase (ATGL) and hormone-sensitive lipase (HSL) located on the LDs membrane possess LC3 protein-binding LC3-interacting region (LIR) motifs. This interaction enables the binding of LDs to the autophagosomal membrane, suggesting the potential of ATGL and HSL as receptors for the specific recognition of lipid droplets. Moreover, ATGL-catalyzed lipolysis may facilitate lipophagy. The degradation mechanism of autophagy for intact organelles is size-dependent, relying on the size of the substrate. Lipolysis aids in adjusting the size of LDs, with liver ATGL preferentially targeting the decomposition of larger-diameter LDs (220 nm–700 nm), generating smaller LDs for engulfment by autophagic vesicles [30]. Additionally, ATGL may activate hepatic lipophagy through the silent information regulator 1 (SIRT1) pathway. The overexpression of ATGL in *mouse* livers results in an increased expression of autophagy-related proteins such as ATG5, LAMP1, and LC3II, enhancing lipophagic activity. However, this effect is suppressed upon *SIRT1* knockout [31].

Furthermore, ATGL-mediated lipolysis requires the participation of lipophagy, as a single mutation in the LIR motif on ATGL can disrupt its lipolytic function [32]. In liver tissues overexpressing ATGL, there is an elevation in LC3II protein levels, a decrease in p62 protein content, and an enhancement of autophagic flux. However, inhibiting the expression of autophagy-related proteins counteracts the beneficial effects of overexpressing ATGL on lipid degradation [31,33]. This suggests that LC3 likely serves as a platform supporting ATGL in exerting its lipolytic function. Another study reveals a potential cooperative relationship between lipophagy and lipolysis. β-adrenaline is a hormone secreted during exercise and mental stress, and the β-adrenaline-β-adrenergic receptor (ADRB2)-lipase system is the main pathway for activating lipolysis. However, after knocking out autophagy genes such as *Rab7* and *Atg5* in cells, the lipolytic effect activated by the ADRB2 pathway is inhibited. Conversely, the overexpression of ADRB2 increases the formation of autophagic lysosomes, enhances autophagic flux, and increases the labeling of LDs by lysosomes, resulting in reduced lipid content [34]. This highlights the crucial role of autophagy in lipolysis.

In summary, ATGL may not only act as a receptor for LD recognition in lipophagy but may also be a key component in the entire signaling cascade. The complementary interactions between lipolysis and lipophagy warrant further investigation.

#### 2.2.2. Molecular Chaperone-Mediated Lipophagy Pathway

CMA is a lysosomal-degradation process mediated by molecular chaperones, targeting specific motif proteins. The specific steps involve the recognition of cytoplasmic proteins containing KFERQ-like sequences by the heat-shock cognate 71 kDa protein (HSC70/HSPA8), forming a substrate–chaperone complex. Subsequently, this complex binds to lysosome-associated membrane protein type 2A (LAMP2A) on the lysosomal membrane. The chaperone complex then alters the substrate conformation, forming the CMA translocation complex. Lysosomal HSC70 (lys-HSC70) then regulates substrate translocation, facilitating the degradation of the substrate by lysosomal proteases. Following degradation, LAMP2A is released from the translocation complex for recycling [35]. Approximately 30% of cytoplasmic proteins contain KFERQ domains, including key enzymes involved in metabolic pathways [36], suggesting that CMA may play a crucial role in regulating cellular metabolism.

The PAT protein family consists of specific proteins located on the membrane of LDs. This family includes perilipin, adipophilin (ADRP), the tail interacting protein of 47 kDa (TIP47), S3-12 protein, and OXPAT, collectively named PLIN1-5. PAT family proteins, which play crucial roles in regulating LD synthesis, degradation, and maintaining LD stability. Among them, PLIN2 and PLIN3 proteins are closely associated with LDs and may function in segregating TG within LDs and providing a protective barrier against lipid degradation in LDs [37].

In fact, PLIN2 and PLIN3 are also important target proteins of molecular chaperone-mediated lipid-droplet autophagy, with their spatial structures containing CMA recognition sequences (KFERQ). Under stress conditions, PLIN2 and PLIN3, with their KFERQ-like sequences, are recognized by HSC70, facilitating their translocation to lysosomes for degradation. This alleviates the restriction on lipid-droplet breakdown, allowing autophagic machinery or lipases direct contact with neutral lipids inside LDs, promoting subsequent lipophagy or lipolysis [37] (as shown in Figure 2). In contrast, hindering the HSP70 recognition of PLINs or knocking out *LAMP2A* results in reduced contact between lysosomes and lipid droplets and increased lipid-droplet accumulation in liver cells [38,39], suggesting that CMA might be a prerequisite for lipolysis and macrolipophagy.

#### 2.2.3. Microphagy Pathway of Lipid Droplets

Microphagy refers to the lysosome-mediated targeted degradation of cargo without the involvement of autophagic intermediates. Schulze’s team, utilizing electron microscopy and immunofluorescence techniques, discovered a direct connection between LDs and lysosomes. Proteins and lipids on LDs can be directly transferred to lysosomes, and this lipid transfer is unaffected by the inhibition of macroautophagy and CMA [40]. This suggests that LDs may directly regulate lipid levels in cells through dynamic interactions with lysosomes. However, the detailed mechanisms involved in this process require further investigation.

## 3. NAFLD and Lipophagy

### 3.1. Liver Metabolic Characteristics of NAFLD

NAFLD represents hepatic manifestations of metabolic syndrome, involving mechanisms such as hepatic steatosis, inflammatory cascades, hepatocellular apoptosis, and fibrosis. Hepatic steatosis is the fundamental characteristic of NAFLD, primarily characterized by aberrations in lipid-metabolism pathways including uptake, synthesis, breakdown, oxidation, and efflux within hepatocytes, liver sinusoidal endothelial cells (LSECs), Kupffer cells, and hepatic stellate cells (HSCs). This disrupts the normal turnover process of lipids, leading to the excessive deposition of TG within hepatic-tissue cells [41].

Lipid droplets serve as the primary storage form of TG in the liver. In pathological conditions, an excessive accumulation of TG within cells leads to increased biogenesis and an enlarged area of LDs, the inhibition of breakdown pathways, and the impediment of TG efflux in the form of very low-density lipoproteins (VLDL), resulting in a massive accumulation of lipid droplets in the liver. Clinically, liver fat deposition exceeding 5% is commonly used as a criterion for diagnosing NAFLD [41]. If hepatic lipid deposition persists without relief, it typically leads to a series of issues including widespread inflammatory responses, hepatocyte death, and fibrosis, further exacerbating the disease progression towards NASH and HCC.

### 3.2. Lipophagy Involvement in NAFLD Development

Lipophagy plays a significant role in regulating lipid turnover and is crucial in NAFLD characterized by hepatic steatosis. Dysregulated lipophagy can lead to lipid accumulation in the liver and induce hepatic steatosis. The specific knockout of the *Atg7* gene in *mouse* livers inhibits cellular lipophagy, resulting in the accumulation of lipid droplets and reduced VLDL secretion [6]. Studies from cellular, animal, and human models suggest that excess nutrition-induced hepatic lipid deposition may inhibit autophagosome formation [7], autophagosome–lysosome fusion [42], lysosomal degradation [43], and disrupt normal lipophagy processes, exacerbating ectopic lipid deposition in the liver [44], thus worsening the condition.

In fact, in addition to NAFL, lipophagy is also closely associated with two other pathological mechanisms of NAFLD, namely, inflammatory responses and cellular apoptosis [7,45,46]. Further investigation reveals that this may be related to the activation of lipophagy programs in different cell types within hepatic tissues. During the early stages of NAFLD (NAFL and NASH), enhanced lipophagy helps clear excess lipid droplets from hepatocytes, reduce apoptotic hepatocyte numbers, and improve inflammatory responses [47]. Lipophagy products such as FFA also provide ample energy substrates for liver-derived macrophages and bone marrow-derived macrophages (BMDMs), promoting their polarization towards M2 anti-inflammatory macrophages, suppressing M1 pro-inflammatory cell polarization, and reducing liver inflammation levels [48,49]. However, studies also show that defective autophagy in liver sinusoidal endothelial cells (LSECs) exacerbates liver fibrosis and inflammation in NASH patients [50]. In contrast, in studies related to hepatic stellate cells (HSCs), it has been found that the activation of lipophagy promotes the transformation of HSCs into myofibroblasts (MFBs). This transformation is characterized by the upregulation of extracellular matrix (ECM) components such as α-smooth muscle actin (α-SMA) and types I and III collagen fibers (Collagen I/III). These changes exacerbate liver fibrosis and worsen the condition. If the disease progresses to the stage of HCC, the breakdown products of lipophagy would provide ample energy support for liver cancer cells, leading to the further deterioration of the condition [51,52]. Therefore, the above research findings suggest that lipophagy is closely associated with pathological features that promote the development of NAFLD, including hepatic lipid deposition, inflammation, and fibrosis (as shown in Figure 3). When exploring the relationship between lipophagy and NAFLD development, it is necessary to consider both spatial (specific roles of lipophagy in different cells) and temporal (disease progression stages) aspects to fully understand the impact of lipophagy on the disease.

## 4. Exercise Inhibits the Onset and Progression of NAFLD through Lipophagy

The occurrence of NAFLD is closely associated with adverse lifestyle habits, with sedentary behavior being a significant contributor to the development of the disease. Individuals, especially children and adults with low physical activity levels, are more susceptible to NAFLD [53], while regular exercise training has been shown to prevent the onset and progression of NAFLD [54]. Studies indicate that exercise plays a role in inhibiting hepatic fat accumulation, alleviating hepatic steatosis [55,56], reducing liver inflammation, and mitigating fibrotic processes [57]. However, the detailed molecular mechanisms underlying these effects are not yet fully understood.

As an effective means to enhance metabolic pathways, exercise transforms the substance/energy metabolism of the body from a state of “disorder” to “order.” This is manifested by disrupting the inherent energy metabolism status of cells during exercise, which, through the coordination of the nervous system and endocrine organs, mobilizes the transport, distribution, and uptake of nutrients within the body. This integration of numerous metabolic pathways within cells aims to meet the cellular requirements for materials and energy. Lipophagy, an important autophagic process regulating lipid turnover within cells, is likely to be involved in the regulation of lipid and energy metabolism in the exercised liver.

### 4.1. Lipophagy Safeguards the Dynamic Lipid Turnover in the Liver during Exercise

#### 4.1.1. Lipid-Droplet Autophagy Directly Contributes to Energy Supply during Exercise

To avoid the toxic effects of lipids, the content and distribution of lipid molecules within cells are precisely regulated. Lipid droplets are isolated lipid pools within cells, undergoing continuous biogenesis and degradation processes (lipid-droplet formation, expansion, and shrinkage) in response to cellular lipid content and energy demands. This lipid-turnover process effectively reflects the cellular response to metabolic demands [14]. Exercise, especially prolonged moderate-intensity exercise, relies on TG within lipid droplets for energy supply. The breakdown of TG releases fatty acids, which can rapidly enter the mitochondria for β-oxidation to generate energy [58]. As a major form of lipid droplet degradation, lipophagy may play a crucial role in mobilizing lipid droplets for energy supply and meeting energy demands [6].

In a study by Li et al., human subjects underwent 12 weeks of aerobic training, resulting in significantly enhanced insulin sensitivity. The researchers also observed that the density, diameter, and volume of the lipid droplets under the sarcolemma were significantly reduced, and a large number of lipid droplets were wrapped in autophagic lysosomes [59]. This suggests that lipophagy may participate in the adaptive remodeling of cellular lipid-droplet dynamics induced by long-term exercise intervention. In an experiment by Gunadi et al. (2020), 8 weeks of exercise training significantly reduced hepatic lipid-droplet content and increased the expression of autophagy-related proteins LC3II/I, Beclin1, Atg5 and the carnitine palmitoyltransferase 1A (*CPT-1A*) gene. Furthermore, the study revealed an intensity-dependent regulation of lipid-droplet content and lipophagy flux in response to aerobic exercise. Compared to *mice* subjected to low-intensity exercise, those undergoing moderate- to high-intensity exercise exhibited a greater depletion of hepatic lipid droplets and higher lipophagy flux [60]. These results suggest that the exercise-induced lipophagy degradation of hepatic lipid droplets may be closely related to the cellular energy demand, primarily driven by fatty-acid oxidation.

Additionally, studies have shown that the expression of hepatic autophagy genes (*Ulk1*, *Becn1*, *Atg5*, *Map1lc3b*, *Sqstm1*) and the mitochondrial biogenesis gene (*Ppargc1a*) is influenced by exercise patterns (aerobic endurance exercise, aerobic exhaustion exercise, resistance exercise, and aerobic endurance combined with resistance exercise). In contrast, aerobic endurance exercise and aerobic exhaustion exercise, which have higher energy demands, can significantly increase hepatic autophagic flux [61]. Unfortunately, this study did not investigate changes in lipid-droplet dynamics or lipid metabolism-related alterations; thus, whether the modulation of lipid-droplet autophagy is associated with exercise patterns remains undisclosed.

#### 4.1.2. Coordination of Lipid-Droplet Autophagy in Regulating Exercise-Induced Energy Provision and Storage

Lipophagy not only participates in cellular energy supply during exercise, primarily using fatty acids as the main substrate, but is also closely associated with exercise-induced lipid-droplet biogenesis. Lipid-droplet degradation generates lipid components used in the synthesis of membrane constituents such as phospholipids, contributing to the expansion of cell and organelle membranes. These lipid components can also be utilized to resynthesize TG stored in newly formed lipid droplets [61]. Inhibiting hepatic autophagy through genetic means leads to defects in lipid-droplet biogenesis [62,63]. Therefore, lipophagy provides crucial support for the self-renewal metabolic mechanism of cellular lipid-droplet biogenesis. A study by La Fuente et al. (2019) [64] demonstrated that 8 weeks of regular exercise significantly increased the number of hepatic lipid droplets in *mice* fed a normal diet, along with increased expression of LC3II/I, SREBP1c, and mTOR downstream p-p70S6K/p70S6K, and decreased p62 expression. An acute exercise study also observed that moderate-intensity treadmill exercise increased the expression of autophagy-related proteins LC3II, p-AMPK, LAMP2, TFEB, reduced p62 expression, and activated the Akt-mTOR synthetic pathway [65]. It is evident that exercise-induced lipophagy may be associated with increased lipid-droplet synthesis, although the specific mechanisms remain unclear.

It is noteworthy that the degradation of lipid droplets through the autophagy pathway leads to the release of a significant amount of fatty acids directly into the cytoplasm [66]. This process helps maintain the availability of the endogenous free fatty-acid pool within the cell but poses significant challenges to the cell: how to appropriately allocate the flow of fatty acids entering the mitochondria (to meet cellular energy demands while avoiding excess reactive oxygen species generation through fatty acid β-oxidation) [67], and how to coordinate the handling of excess FFA in the cytoplasm (to prevent cellular toxicity). The findings of Nguyen et al. (2017) [68] suggest that lipid-droplet biogenesis is a common response of cells to high-throughput autophagy, possibly representing an active manifestation of cellular control over FFA. Lipid droplets can isolate a portion of the fatty acids produced through autophagy, preventing lipid overload in mitochondria and cells. Inhibiting lipid-droplet synthesis by suppressing the DGAT1 function results in decreased mitochondrial oxidative respiratory capacity, membrane potential abnormalities, and increased cell death [68]. This indicates that lipid-droplet autophagy not only meets the cell’s demand for FFA but also facilitates the efficient processing of cytoplasmic lipid molecules (isolating fatty acids in the form of lipid droplets). Therefore, from the perspective of lipid metabolism, lipid-droplet autophagy-driven efficient lipid turnover may contribute to explaining the seemingly contradictory phenomenon of increased lipid droplet content and enhanced metabolic capacity in the muscles of elite endurance athletes, known as the “athlete’s paradox” [69]. Excessive lipid-droplet reserves and efficient lipid mobilization can provide more abundant energy support for athletic tissues. Exercise-induced adaptive increase in liver lipid droplets ensures an adequate supply of energy substances from the liver during prolonged endurance exercise, meeting energy demands and enhancing athletic performance [64].

In summary, lipid-droplet autophagy may play a crucial pivotal role in the lipid turnover of a healthy liver during exercise, endowing the liver with elasticity in lipid metabolism. Autophagy breaks down lipid droplets to generate fatty acids, meeting the cell’s demand for fatty acids during exercise, while also promoting the efficient processing of cytoplasmic lipids, avoiding cellular toxicity. This may likely result in the seemingly contradictory phenomenon of tissue accumulating lipid droplets beyond physiological levels accompanied by an enhanced metabolism. However, current evidence is insufficient to clearly explain the specific connections and mechanisms between exercise-induced lipid-droplet autophagy and lipid turnover, requiring further elucidation.

### 4.2. Exercise Improves Hepatic Fat Deposition through Lipophags

Research has found that various forms of exercise, including aerobic endurance exercise, high-intensity interval training, and resistance exercise, effectively reduce weight, lower lipid levels, and alleviate hepatic fat content in patients with NAFLD [9,70]. Abnormal hepatic lipid turnover due to diet or lifestyle-induced factors is a primary cause of hepatic lipid-droplet deposition, leading to hepatic fat degeneration, further disrupting lipophagy processes, aggravating fat accumulation, and compromising normal liver function. Therefore, regulating hepatic lipophagy may provide potential prevention and treatment for hepatic fat deposition.

It has been reported that hepatic lipophagy is suppressed in high-fat diet-fed *mice*, resulting in massive lipid-droplet accumulation. Rosa-caldwell et al. (2017) [71] found that four weeks of wheel running significantly increased LC3II/I expression, downregulated p62 expression, and reduced abnormal lipid deposition in the livers of high-fat diet-fed *mice*. Sixteen weeks of moderate-intensity swimming exercise also activated the AMPK-SIRT1 pathway in the livers of high-fat diet-fed *mice*, upregulated the expression of ULK1, LC3II/I, and LAMP1, increased the colocalization level of LD with LC3 and LAMP1, promoting lipid-droplet degradation through lipophagy and inhibiting abnormal lipid-droplet accumulation in the livers of high-fat diet-fed *mice* [72]. In a study of combined exercise and dietary intervention, it was found that exercise and fasting, through the AMPK/ULK1 pathway and Akt/mTOR/ULK1 pathway, respectively, activated lipophagy, improving hepatic fat deposition. These studies suggest that aerobic training can reduce hepatic lipid deposition by activating lipophagy. Other research indicates that 12 weeks of moderate-intensity swimming training, by regulating the FABP1 pathway in the livers of high-fat diet-fed *mice*, restored lysosomal protease hydrolysis and acidification functions, enhanced hepatic lipophagy flux, and reduced lipid deposition [73]. Guo et al. (2020) [74] also found similar research conclusions; three weeks of treadmill training enhanced lipophagy by inhibiting the p-AKT/mTOR pathway in high-fat diet-fed SD rats, alleviating lysosomal membrane permeability, preventing tissue proteinase B release, and mitigating lipid accumulation. This indicates that exercise training can maintain lysosomal stability and promote normal lipophagy flow. In addition, research shows that simulating knee-joint loading can upregulate LC3II/I protein levels in the livers of ovariectomized female NAFLD *mice*, reduce p62 protein expression, and decrease hepatic lipid-droplet deposition [13].

In conclusion, exercise training can regulate the lipophagy process by promoting intervention initiation, maintaining lipophagy flux, etc., improving hepatic fat deposition. Moreover, different aerobic intervention methods such as treadmill running, voluntary wheel running, swimming, or simulated exercises can reduce hepatic lipid deposition through the lipophagy process. However, there is currently a lack of exploration into the effects of different forms (such as resistance exercise, high-intensity interval training) and intensities of exercise interventions on hepatic lipophagy and lipid-droplet deposition characterization in NAFLD.

### 4.3. Exercise Alleviates Liver Inflammatory Response and Fibrosis via Lipophagy

The progression of NAFLD often involves abnormal hepatocyte swelling, inflammatory outbreaks, fibrosis, and even the formation of tumor nodules in the liver, marking the transition of the disease into NASH or even HCC stages. Research has found that 16 weeks of treadmill exercise can significantly reduce serum ALT levels in HFD *mice*, downregulate hepatic tumor necrosis factor (TNF-α) and α-smooth muscle actin expression, reduce macrophage infiltration, and inhibit hepatic stellate cell activation, thereby alleviating NAFLD scores [57]. This suggests that exercise may alleviate liver inflammation, reduce liver damage, improve liver fibrosis, and inhibit disease progression.

Guarino et al. (2020) [75] found that exercise alleviates the exacerbation of NAFLD and is related to lipophage regulation. Feeding *mice* a choline-deficient high-fat diet (CD-HFD) for 12 weeks leads to severe hepatic steatosis, and continuing this diet for an additional 8 weeks results in significant hepatocyte swelling, fibrosis, and even hepatocellular adenomas. However, if aerobic treadmill exercise is performed concurrently with CD-HFD feeding for 8 weeks, it significantly increases the expression of p-AMPK, LC3II/I, and ATG5 in the livers of CD-HFD *mice*, reduces TNFα expression, decreases hepatic lipid content, inhibits hepatocyte swelling, and alleviates inflammation and fibrosis. Notably, after 8 weeks of aerobic training, the tumor incidence in CD-HFD *mice* decreases to 70%. This suggests that exercise may potentially alleviate NAFLD progression to NASH or even HCC by activating hepatic lipophagy to suppress inflammation and fibrosis, although further evidence is needed. In studies of exercise combined with dietary intervention, it was found that 14 months of treadmill training or treadmill training combined with DHA feeding can enhance liver LC3II/I and Atg7 expression in elderly DIO *mice* and reduce p62 protein and pro-inflammatory gene expression such as *MCP1*, *IL6*, *TNFα*, and *TLR4* [76], thus alleviating NAFLD progression. Additionally, research has shown that 8 weeks of aerobic training significantly reduces liver lipid content in NAFLD *mice*, decreases the expression of inflammatory factors such as TNF-α, IL-10, MCP-1, and IL-6, and is accompanied by activation of the AMPK-PPARα pathway [77]. PPARα can regulate the transcriptional output of many lipophagy-related genes by binding to the promoters of numerous autophagy genes [78]. This also suggests that lipophagy may be one of the potential pathways through which exercise improves liver inflammation.

### 4.4. The Ambiguous Role of Lipophagy in the Exercise-Induced Improvement of NAFLD Progression

Although, the above research results indicate that the activation of lipophagy during exercise may help prevent the development of NAFLD. Moreover, cellular-level studies have indicated that exercise mimetics can inhibit hepatic lipid-droplet accumulation by activating lipophagy. However, whether exercise can regulate lipophagy in hepatic resident macrophages and hepatic stellate cells, and the physiological implications of this process on the development of NAFLD, remain elusive. Resident hepatic macrophages and hepatic stellate cells are the main hepatic cells that contribute to inflammation and fibrosis. Studies have found that 8 weeks of downhill running can activate M2 macrophages in the livers of DIO *mice*, inhibit M1 macrophage polarization, alleviate inflammation, and prevent disease progression [79], which seems to be consistent with the benefits of activating lipophagy in hepatic macrophages [80]. It is speculated that exercise alleviating liver inflammation may be related to enhanced lipophagy in macrophages. In contrast, in studies of hepatic stellate cell regulation of liver fibrosis, the regulatory effects of exercise and lipophagy seem to have opposite effects. Hernandez-Gea et al. (2012) [80] found that enhancing lipophagy activates hepatic stellate cells in *mice* or humans, exacerbating liver fibrosis; inhibiting ATGs such as Atg5 and Atg7 can alleviate liver fibrosis, indicating that reducing lipophagy may be a way to inhibit HSC activation. However, research has found that 12 weeks of moderate-intensity exercise can reduce the mRNA expression of transforming growth factor-β (TGF-β) and α-smooth muscle actin in NASH rats, increase the expression of cysteine-rich angiogenic inducer 61 (Cyr61), inhibit HSC activity, and reduce liver fibrosis scores [81]. This suggests that exercise can inhibit HSC activation and reduce liver fibrosis, but this seems to contradict the effects of activating HSC lipophagy. It indicates that exercise inhibiting HSC activation may be related to the negative regulation of lipophagy. Therefore, based on the above evidence, it is speculated that the regulatory effects of exercise training on hepatic lipophagy may not be consistent during different stages of disease development, and it may be necessary to explore the lipophagy regulation mechanisms of different cell types in the liver.

### 4.5. Exercise Factors as Potential Pharmacological Targets for Regulating Hepatic Lipophagy

Exercise/muscle contraction is a classical physiological stimulus characterized by muscle movement, which balances the body’s nutrition and energy status by regulating the working state of various cellular components to meet the body’s needs. The liver is not a direct target organ of exercise, but hepatic lipophagy is influenced by exercise. Exercise can affect hepatic lipid metabolism through lipophagy regulation, thereby alleviating the development of NAFLD. However, the detailed mechanisms remain unclear.

In fact, the vigorous metabolic demands during exercise require the involvement of various organs throughout the body, stimulating numerous endogenous cellular factors to enter the bloodstream, regulate body metabolism, repair organ damage, and alleviate organ aging [82].

#### 4.5.1. Irisin

Irisin is the active form of fibronectin type III domain-containing 5 (FNDC5), a membrane protein component. During exercise, the extracellular end of FNDC5 is cleaved and regulates the metabolic levels of other tissues and organs through the bloodstream [83]. Studies have found that FNDC5 can activate hepatic lipid scavengers via the AMPK/mTORC1 pathway, promote fatty acid oxidation, and improve lipid accumulation [84]. Additionally, nicotinamide adenine dinucleotide (NAD+), a key coenzyme in regulating energy metabolism during exercise, significantly increases serum Irisin levels in *mice* and humans with nicotinamide riboside (NR) treatment, improving hepatic lipid degeneration in NAFLD *mice*. However, NR treatment has no significant improvement effect on *FNDC5*−/− *mice* with lipid degeneration [85]. This suggests that FNDC5 secreted by the liver may activate lipid scavengers in response to the energy demands of exercise, improving hepatic lipid degeneration.

#### 4.5.2. Fibroblast Growth Factor 21

Fibroblast Growth Factor 21 (FGF21) is a myokine secreted by the liver, muscle, and adipocytes. Studies have found that mRNA expression of FGF21 in the liver of NASH rats increases after exercise training, leading to elevated serum levels of FGF21 and improved hepatic oxidative stress and glucose metabolism [86]. Geng et al. (2020) [87] also found that 7 weeks of aerobic exercise promotes hepatic secretion of FGF21, improving glucose tolerance in NASH *mice*. Furthermore, research has shown that 16 weeks of treadmill exercise significantly enhances hepatic FGF21 expression, suppresses excessive lipid synthesis, and alleviates lipid deposition [88]. These findings suggest that FGF21 may be a key factor in the aerobic exercise-induced improvement of glucose and lipid metabolism, though the mechanisms remain unclear. However, recent studies indicate that FGF21 may improve glucose and lipid metabolism dysregulation and alleviate NAFLD progression by activating and regulating lipophagy. Eight weeks of aerobic exercise significantly enhances hepatic FGF21 expression, activates lipophagy via the AMPK-ULK1 pathway, improves hepatic lipid deposition, and alleviates liver damage [89]. Additionally, in studies by Obydah et al. (2021) [90], 8 weeks of aerobic exercise significantly increased hepatic and serum FGF21 levels, accompanied by increased hepatic LC3II/I ratio, decreased p62 expression, and reduced hepatic lipid deposition. Indeed, under fasting conditions, FGF21 can also promote autophagy by phosphorylating Jumonji-D3 (JMJD3/KDM6B) histone demethylase at the thr1044 site via PKA, facilitating its binding to PPARα, thereby transcriptionally regulating *TFEB*, *ATG7*, and other autophagy-related gene expressions, activating hepatic lipophagy and improving fat deposition [91]. In summary, FGF21 may be a key factor in the exercise-mediated regulation of the liver, but the specific mechanisms remain incompletely understood and warrant further investigation.

## 5. Summary and Research Limitations

Lipophagy is a selective autophagic mechanism. Under conditions of energy deficiency and nutrient deprivation, lipid droplets can be degraded through three pathways: macroautophagy, microautophagy, and CMA. Lipophagy promotes lipid-droplet breakdown, mobilizes lipid metabolism within cells, facilitates lipid turnover at the cellular/tissue/organ levels, and maintains cellular lipid metabolism flexibility. However, unhealthy lifestyles can disrupt normal hepatic cellular lipophagy processes, leading to impaired lipid turnover, hepatic lipid deposition, inflammation, fibrosis, hepatocyte apoptosis, and other pathological changes, driving the progression of NAFLD. Exercise is one of the important means to improve NAFLD, potentially by regulating hepatic lipophagy, promoting hepatic lipid turnover, improving hepatic steatosis, and alleviating inflammatory responses and fibrosis processes, thereby ameliorating NAFLD. Unfortunately, apart from IRISIN and FGF21, which have been under drug development for modulating hepatic lipophagy, other exercise factors that could regulate hepatic lipophagy have not yet been discovered, thus requiring further investigation.

However, to date, research on hepatic lipophagy in nonalcoholic fatty liver disease (NAFLD) has been relatively focused on the stage of hepatic steatosis, while studies on disease progression (hepatitis, liver fibrosis, liver cancer) remain relatively scarce. Additionally, there is currently insufficient evidence at the cellular or animal level to reveal the impact of hepatic non-parenchymal cells (such as hepatic macrophages and hepatic stellate cells) on NAFLD through lipophagy. Therefore, it is currently difficult to determine whether lipophagy is beneficial or detrimental in the later stages of NAFLD. Furthermore, although exercise has been shown to have beneficial effects on the late-stage development of NAFLD, the contributions of hepatic parenchymal and non-parenchymal cell lipophagy, as well as the specific lipophagy regulatory mechanisms in different cells, remain unclear. Moreover, identifying key molecular markers of exercise-regulated lipophagy in the occurrence and development of NAFLD could provide a ‘molecular ruler’ to quantify exercise standards, further aiding in the prevention and treatment of NAFLD. Additionally, the development of small molecule drugs targeting lipid-droplet autophagic degradation (autophagosome-targeting compounds, ATTEC), as enriched by Fu et al. (2021) [92] has expanded the pharmacological means to improve hepatic lipid deposition. However, the clinical application and efficacy of these drugs require further validation over time.

## Figures and Tables

**Figure 1 ijms-25-03153-f001:**
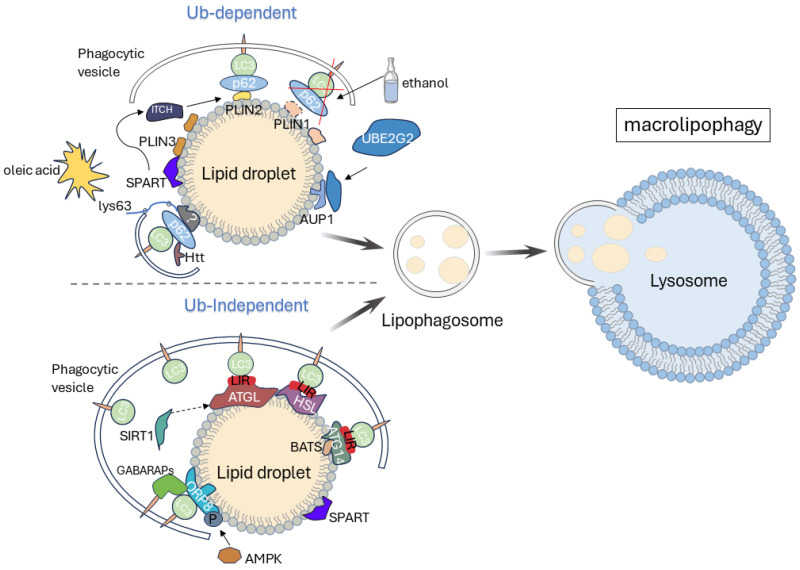
Macrolipophagy pathway (this figure is adapted from [19]). The mechanisms involved in ubiquitin-dependent macrolipophagy include the following: (1) under oleic acid exposure conditions, the C-terminal domain of SPART binds to PLIN3, serving as an adaptor protein to recruit a ubiquitin ligase homologous to the C-terminus of E6-AP. The ubiquitin ligase ITCH/AIP4 can be activated by SPART and recruited to LD, promoting the binding of PLIN2 to the autophagic receptor p62; (2) Htt can interact with p62, facilitating the connection between p62 and LC3 and substrates marked with lys63 ubiquitin, thereby enhancing the contact between LDs and phagosomes; (3) the LD-membrane protein AUP1 can recruit and bind to the E2 ubiquitin-conjugating enzyme UBE2G2. Furthermore, under ethanol exposure conditions, the loss of PLIN1 weakens the interaction between LC3, p62, and LD, which may be associated with the inhibition of ubiquitin-dependent macrolipophagy. The mechanisms involved in ubiquitin-independent macrolipophagy include the following: (1) under starvation conditions, ORP8 is phosphorylated via AMPK, enhancing its affinity for LC3/GABARAPs; (2) ATG14 is localized to the LD membrane via its own BATS domain and recruits LC3II through the LIR motif; (3) ATGL and HSL interact with LC3 through the LIR motif, and ATGL may activate lipophagy through the SIRT1 pathway. Additionally, SPART is mainly involved in mediating neuronal LD autophagic degradation.

**Figure 2 ijms-25-03153-f002:**
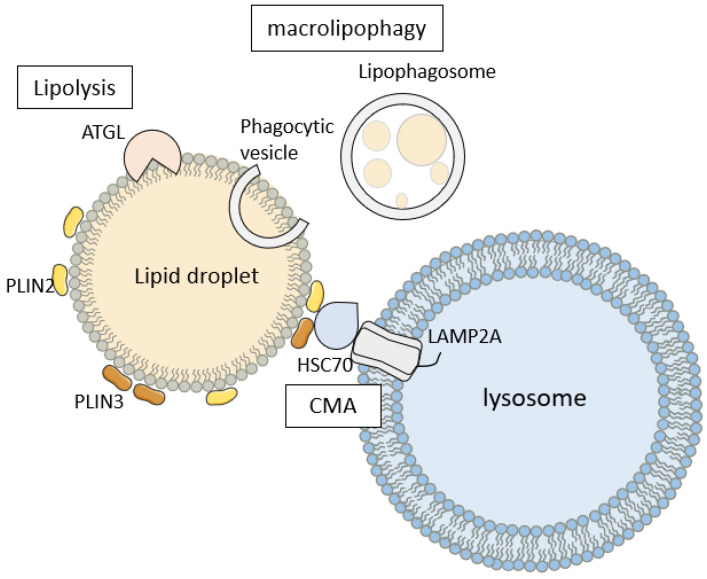
Molecular partner-mediated lipophagy pathway. Under stress conditions, the KFERQ domains on PLIN2 and PLIN3 are recognized by HSC70, forming substrate–chaperone complexes. These complexes bind to LAMP2A on the lysosomal membrane, where the chaperone complex alters the conformation of the substrate, forming CMA translocation complexes. Subsequently, lys-HSC70 mediates substrate translocation, promoting the degradation of PLIN2 and PLIN3 by lysosomal proteases. Losing the protection of PLIN2 and PLIN3 makes the neutral lipids in the lipid droplets more susceptible to being accessed by cytoplasmic ATGL or phagocytic vesicles, triggering subsequent lipolysis or lipophagy.

**Figure 3 ijms-25-03153-f003:**
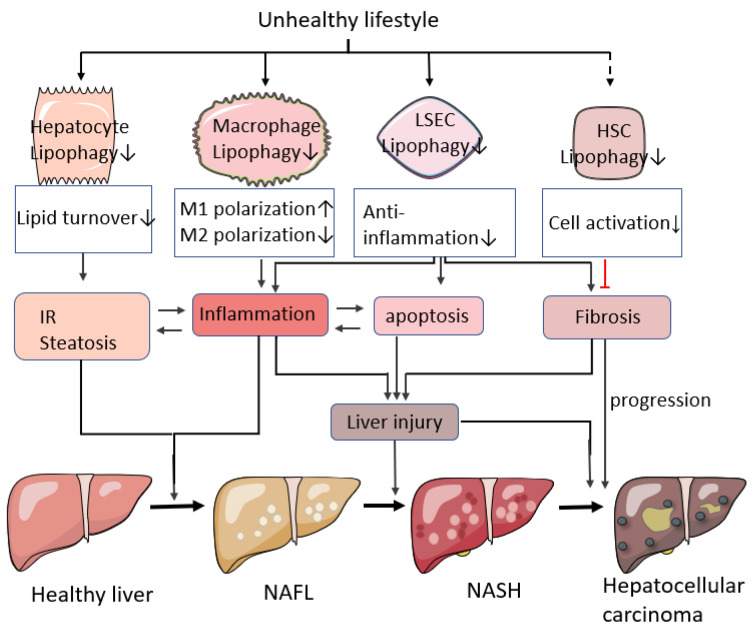
Adverse lifestyle habits disrupt cellular lipophagy in liver tissue, affecting the onset and progression of NAFLD. “

” means enhanced; “

” means weakened; “
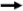
” stands for promotion; “
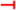
” indicates inhibition; A dashed line means it is not clear.

## Data Availability

Not applicable.

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
