# Peer review of "Research Progress on Lipophagy-Mediated Exercise Intervention in Non-Alcoholic Fatty Liver Disease"

_ijms, 2024, doi:10.3390/ijms25063153_

Round 1

Reviewer 1 Report

Comments and Suggestions for Authors

The paper explores the role of lipophagy, a cellular pathway involved in the degradation of lipid droplets, in the context of non-alcoholic fatty liver disease (NAFLD) and the potential mechanisms by which exercise training influences this process. Overall, the topic is relevant and timely given the rising prevalence of NAFLD and the importance of understanding its underlying mechanisms and potential interventions.

The authors provide a clear explanation of lipophagy and its relevance to lipid turnover and NAFLD. This clarity is essential for readers who may not be familiar with the topic, making the manuscript accessible to a broader audience. The authors have also appropriately integrated recent research findings to support the topic. This strengthens the argument and highlights the relevance of the topic. 

However, a few suggestions for improvement include:

1. It would be valuable to discuss the potential clinical implications of the findings presented in the manuscript. For example, how could targeting lipophagy pathways pharmacologically or through lifestyle interventions be translated into clinical practice for the prevention or treatment of NAFLD? Addressing these aspects would enhance the relevance and practical utility of the research.

2. It would be beneficial to include a section discussing the limitations of current research in this area and proposing avenues for future investigation. Addressing the gaps in understanding and potential challenges in translating findings into clinical practice would provide a more comprehensive perspective on the topic.

3. The manuscript would benefit from more illustrations and tables. Illustrations are great especially where molecular pathways are described.

Reviewer 2 Report

Comments and Suggestions for Authors

In this review, Xi Li and colleagues analyze the interconnection between lipophagy, NAFLD, and exercise. They also elucidate potential mechanisms through which exercise, via lipophagy, may improve NAFLD pathology. The review is well structured and offers a comprehensive view of NAFLD, lipophagy and the mechanisms activated by exercise to reduce the onset and progression of this disease. However, I have some comments to raise with the authors to improve the quality of the work and make it suitable for publication in the IJMS journal.

Lines 31-39: Considering lipophagy is a form of selective autophagy, in the introduction section, the authors should explain the autophagy process in detail, also referring to recent works in the literature. Examples are given below (https://pubmed.ncbi.nlm.nih.gov/35205361/).

Lines 41-42: The authors mention that exercise reduces ectopic fat deposits in the liver. It is known in the literature that mild exercise leads to an increase in beta-oxidation of fatty acids in the liver. The authors should provide further examples of this intervention and other types of exercise.

Line 102: Explain the names of the factors in full.

Sections 1.2.1.1 and 1.2.1.2 can be made more comprehensible and fluent by inserting pictures of these mechanisms.

Please add a description to figure 1 and the programme from which the image was obtained.

Sections 3.1 and 3.3 are quite long and the reader may get lost understanding it. Please ask the authors to divide it into smaller sections.

Comments on the Quality of English Language

Minor editing of english language
